



# Streamflow data availability in Europe: a detailed dataset of interpolated flow-duration curves

Simone Persiano[1], Alessio Pugliese[1], Alberto Aloe[2], Jon Olav Skøien[2], Attilio Castellarin[1], Alberto Pistocchi[2]

[1]Department of Civil, Chemical, Environmental and Materials Engineering (DICAM), University of Bologna, Bologna, Italy
[2]European Commission, DG Joint Research Centre (JRC), Ispra, Italy

*Correspondence to*: Alberto Pistocchi (alberto.pistocchi@ec.europa.eu)

**Abstract.** For about 24,000 river basins across Europe, we provide a continuous representation of the streamflow regime in terms of empirical flow–duration curves (FDCs), which are key signatures of the hydrological behaviour of a catchment and

are widely used for supporting decisions on water resources management as well as for assessing hydrologic change. In this study, FDCs are estimated by means of the geostatistical procedure termed total negative deviation top-kriging (TNDTK), starting from the empirical FDCs made available by the Joint Research Centre of the European Commission (DG-JRC) for about 3,000 discharge measurement stations across Europe. Consistent with previous studies, TNDTK is shown to provide high accuracy for the entire study area, even if with different degrees of reliability, which varies significantly over the study

area. In order to provide this kind of information site-by-site, together with the estimated FDCs, for each catchment we provide indicators of the accuracy and reliability of the performed large-scale geostatistical prediction. The dataset is freely available at the open access library PANGAEA (Data Publisher for Earth & Environmental Science) at https://doi.pangaea.de/10.1594/PANGAEA.938975 (Persiano et al., 2021).

## 1 Introduction

Over the past decades, the increasing accessibility of global datasets (soil, land-cover, morphology, climate characteristics, satellite-based gridded precipitation, etc.) and the ever-expanding computational capabilities have triggered the development of macro-, continental- and global-scale rainfall-runoff simulation models, which are already state-of-the-art (see e.g. Collischonn et al., 2007; de Paiva et al., 2013). Macro-scale models are getting more and more popular and accurate in terms of average and performance over large areas; their simulated streamflow series, some of which are open-access and freely

distributed, represent a wealth of information for addressing a variety of water problems, such as the streamflow regime prediction in data scarce regions of the world (Pechlivanidis and Arheimer, 2015) and the implementation of large-scale and trans-boundary policies for water resources system management or flood-risk mitigation (de Paiva et al., 2013; Sampson et al., 2015; Falter et al., 2016). Due to the impossibility to perform comprehensive calibrations and validations of such models over the whole modelled regions, local performance can be rather diverse (see e.g. de Paiva et al., 2013; Donnelly et al., 2016),



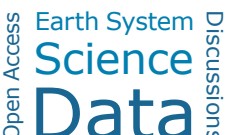

depending on several factors, e.g. the quality of macro-scale input data, the capability of the selected conceptual scheme to accurately represent the dominant hydrological processes locally governing the rainfall-runoff transformation (geological and morphological or climatic and micro-climatic factors).

An empirical characterization of the natural streamflow regime over large areas would be a fundamental piece of information for benchmarking the performance of macro-scale models and for assessing their potential locally. This necessity conflicts

with the availability and accessibility of streamflow observations, which can be limited even in technologically advanced regions of the world.

In this context, along the lines of the study conducted by Castellarin et al. (2018) for the Danube region, this study performs a statistical regionalization of streamflow regimes in Europe, compiling a data-driven benchmark dataset for hydrological models, as well as a data layer to be made available for a broader use. The streamflow regime for each catchment is

characterized in terms of flow-duration curve (FDC), a graphical representation of the frequency (i.e. percentage of time, or duration) with which a given streamflow is equalled or exceeded over an historical period of time at a given river basin (see e.g. Vogel and Fennessey, 1994). Providing a simple and compact view of the historical variability of streamflows, from high flows to low flows, an FDC is a key signature of the hydrological behaviour of a catchment: its shape reflects climate conditions and the hydrogeological characteristics (i.e. size, morphology, permeability) of the catchment itself (see e.g. Castellarin, 2014;

Westerberg et al., 2016). For this reason, FDCs are routinely used for addressing water resources management problems such as hydropower feasibility studies, classification of streamflow regimes, irrigation planning and management, definition of environmental flows, habitat suitability studies (e.g. Vogel and Fennessey, 1995; Yaeger et al., 2012), as well as for assessing hydrologic change at a river cross-section (Kroll et al., 2015; Ceola et al., 2018).

Starting from a compilation of about 3,000 discharge measurement stations across Europe, where streamflow indices and

empirical period-of-record FDCs (i.e. 15 streamflow quantiles) are compiled by the Joint Research Centre of the European Commission (DG-JRC) from the archives of the Global Runoff Data Centre (GRDC), here we perform a geostatistical interpolation of the streamflow regime to provide FDCs estimates for a total of 24,148 elementary catchments for the European region. To this aim, high-quality empirical period-of-record flow-duration curves (POR-FDCs) are interpolated over the stream-network using the geostatistical procedure termed total negative deviation top-kriging (TNDTK; Pugliese et al., 2014,

2016), which has been shown to provide reliable predictions of continuous FDCs at ungauged locations, overcoming the limit of regression models of modelling streamflow quantiles independently of each other (Castellarin et al., 2018; Pugliese et al., 2016), and to be helpful in significantly enhancing macro-scale simulations (Pugliese et al., 2018). In line with Castellarin et al. (2018), together with the estimated FDCs, for each elementary catchment we provide indicators of the accuracy and reliability of the performed large-scale geostatistical prediction.



## 2 Data and methods

### 2.1 Source data and screening

The present study uses a database compiled by the Joint Research Centre of the European Commission (DG-JRC), consisting of 3,138 streamgauges across Europe. For the catchment upstream each gauged station, streamflow indices and several catchment descriptors are available. Streamflow indices are computed from the streamflow time-series observed at each gauge and consist of mean annual flow (MAF, long-term average daily discharge) and 15 streamflow quantiles associated with duration of 1, 5, 10, 20, 30, 40, 50, 60, 70, 75, 80, 90, 95, 97 and 99.7%. The quality of streamflow data is classified as high and low: high-quality data refers to gauging stations with a precise positioning along the stream that are unique in their elementary sub-basin (i.e. portion of basin directly drained by a river stretch, between two confluences, or from the headwater to the first confluence), whereas low-quality data refers to cases in which more streamgauges are present in a single elementary basin, hence potentially affected by imprecise positioning along the stream (see also Castellarin et al., 2018). In particular, 3,004 study catchments are extracted from the original DG-JRC database of 3,138 measuring points by consolidating and merging multiple entries (i.e. in case of streamgauges redundancy for a given location, only the highest quality streamgauge associated with the largest drainage area is retained).

Among the 3,004 selected catchments, empirical MAF values observed at high-quality measuring points show minimum, 1st quartile, median, mean, 3rd quartile and maximum equal to 0.01, 3.03, 9.41, 80.16, 35.38 and 6,378.00 m³/s, in this order. Consistent with Castellarin et al. (2018), based on the values of MAF standardized by catchment area (i.e. unit mean annual flow, MAF/Area) as a function of basin area (see Fig. 1), the 146 measuring points falling outside the interval 0.0015–0.08 m³ s$^{-1}$ km$^{-2}$ are regarded as highly discordant and therefore excluded from further analyses. Among the 2858 retained catchments, Fig. 1 and Fig. 2 highlight the predominance of streamgauges with high-quality data (2484) over the low-quality ones (374, mainly concentrated in the area of the Danube region; see Castellarin et al., 2018).

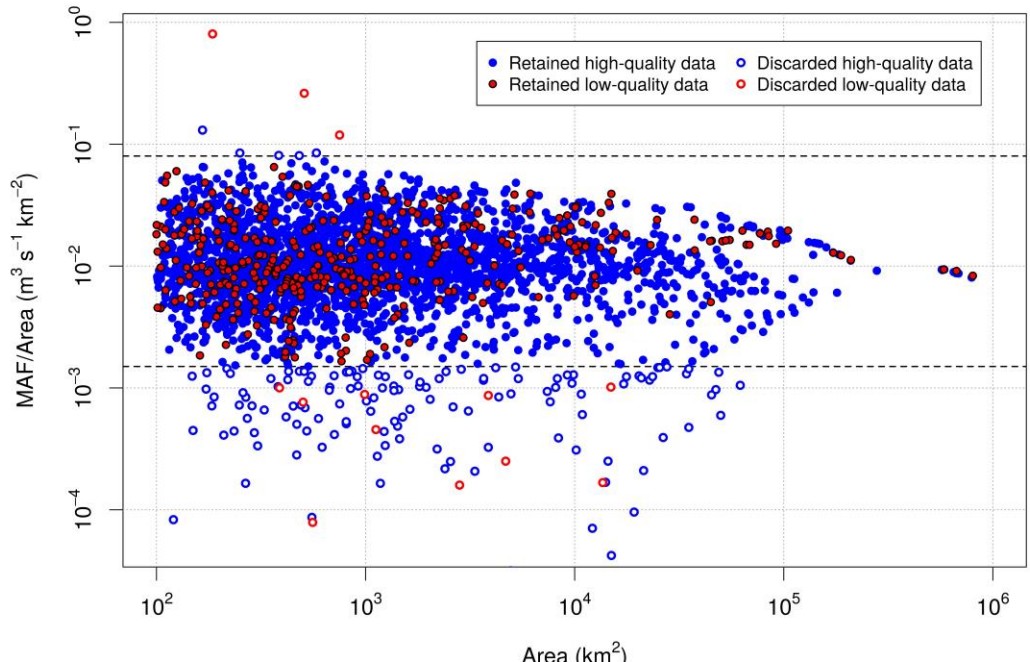

**Figure 1. Unit mean annual flow (MAF/Area) as a function of basin area for the 3,004 study catchments. Streamgauges located outside the interval 0.08–0.0015 m³ s⁻¹ km⁻² (see horizontal dashed lines) are identified as highly discordant and removed from the dataset. Blue and red points are associated with high- and low-quality data, respectively.**

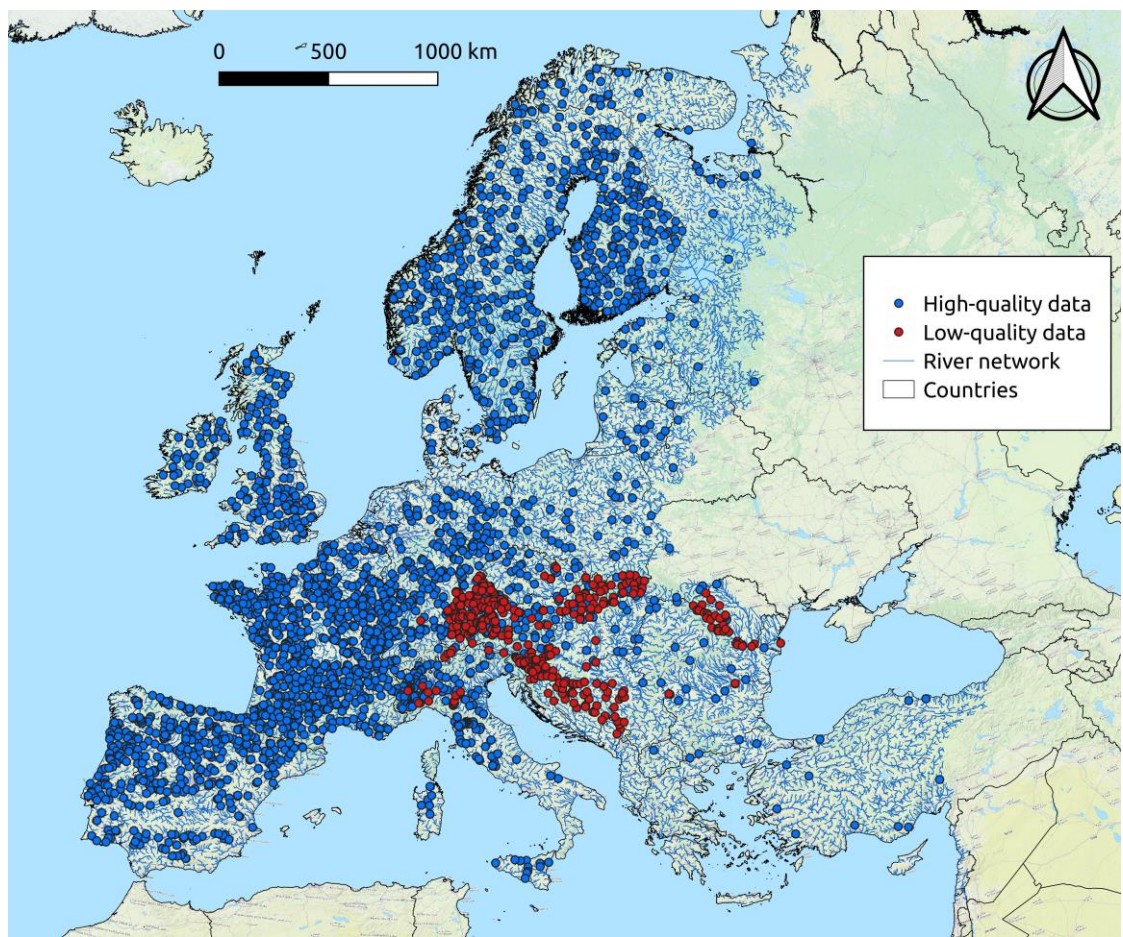

**Figure 2. Streamgauges (2858) retained from the DG-JRC database. Blue and red points refer to high- and low-quality data, respectively.**

Differently from the case study of the Danube region described in Castellarin et al. (2018), for which the analysis was performed twice (first by considering low- and high-quality data combined and then by focusing only on high-quality data), for the European continent we focus only on the 2484 measuring points associated with high-quality data.

Together with the above-mentioned streamgauges, the DG-JRC identifies a layer of 32,960 prediction nodes over the entire European region, for which we perform the prediction of FDCs described herein.

**2.2 Methods: geostatistical interpolation**

Based on what performed for the Danube region in Castellarin et al. (2018), FDCs predictions at the DG-JRC catchments are obtained by applying the geostatistical procedure named total negative deviation top-kriging (TNDTK; Pugliese et al., 2014,





2016). TNDTK is based on top-kriging (Skøien et al., 2006, 2014), a geostatistical tool which is widely used in the literature for predicting streamflow indices (e.g. low-flows, see Castiglioni et al., 2011; Laaha et al., 2014; high-flows and floods, see (Merz et al., 2008; Archfield et al., 2013; Persiano et al., 2021), habitat suitability indices (Ceola et al., 2018), daily streamflow

series (Skøien and Blöschl, 2007; de Lavenne et al., 2016; Farmer, 2016) at ungauged river cross-sections as linear combinations of the empirical information collected at neighbouring gauging stations by accounting for catchment size and nesting structure of the stream network. Specifically, TNDTK (Pugliese et al., 2014, 2016) uses top-kriging in an index-flow framework (Castellarin et al., 2004) for predicting the entire FDC at ungauged sites, ensuring its monotonicity. To this aim, as a first step, the dimensionality of the problem is reduced by standardizing the empirical FDC at site $x$, $\Psi(x, d)$ (where $d$ is a

specific duration), for a reference value (e.g. mean annual flow, MAF), $Q^*(x)$, to yield a dimensionless FDC, $\psi(x, d)$:

$$\psi(x, d) = \frac{\Psi(x, d)}{Q^*(x)} \tag{1}$$

Pugliese et al. (2014) identify an overall point index, named total negative deviation (TND), that effectively summarizes the entire curve: TND is derived by integrating the area between the lower limb of the FDC and the reference streamflow value $Q^*(x)$ (see e.g. Fig. 1 in Pugliese et al., 2014). In particular, TND is used as a regionalized variable to develop site-specific

weighting schemes. The same weights, derived through the solution of the linear kriging system, are used for a batch prediction of the continuous dimensionless FDC, $\hat{\psi}(x_0, d)$, for the ungauged site $x_0$:

$$\hat{\psi}(x_0, d) = \sum_{j=1}^n \lambda_j \, \psi(x_j, d) \quad \forall d \in (0,1) \tag{2}$$

where $\lambda_j$, for $j = 1, \dots, n$, are the weights resulting from the kriging interpolation of TNDs for the $n$ neighbouring gauged catchments; $\psi(x_j, d)$ is the dimensionless empirical FDC at the donor site $x_j$. Equation (2) highlights that the computation of

kriging weights depends on $n$, the number of neighbouring sites on which to base the spatial interpolation.

Once a reliable model (e.g. a regional regression model, or kriging model) for predicting $Q^*$ at the ungauged site $x_0$ (i.e. $\hat{Q}^*(x_0)$) has been set up for the study region, the prediction of the dimensional FDC, $\hat{\Psi}(x_0, d)$, can be obtained as:

$$\hat{\Psi}(x_0, d) = \hat{Q}^*(x_0) \, \hat{\psi}(x_0, d) \quad \forall d \in (0,1) \tag{3}$$

For the sake of brevity, this prediction method is referred to as total negative deviation top-kriging (TNDTK). Additional

details can be found in Pugliese et al. (2014). TNDTK has been shown to provide reliable predictions of FDCs at ungauged sites over large study areas (Castellarin et al., 2018) and to reliably reconstruct natural FDCs at ungauged sites (Ceola et al., 2018).





## 3 Large-scale geostatistical prediction and technical validation

### 3.1 Application of the geostatistical interpolation

In the present study, TNDTK is applied by implementing the R-package "rtop" (Skøien et al., 2014). The mean annual flow (MAF) is chosen as the reference streamflow value $Q^*$ used for standardizing empirical FDCs across the study region and empirical TND values are computed for each empirical dimensionless FDC, standardized by local MAF values. It is worth highlighting that, while computing TND values, the durations of interest are transformed into standard normal variates using the meta-Gaussian transformation, which enhances the representation of the right tail of FDCs, and therefore better

differentiates TND values associated with different empirical curves. Either for MAF or TND interpolations, top-kriging is implemented by following the following steps: (1) the binned sample variogram is calculated, (2) the 5-parameter theoretical variogram (i.e. a "modified" exponential model, which combines an exponential model with a fractal model, see details in Skøien et al., (2006) is then regressed against the sample variogram through a weighted least squares (WLS) regression method (Cressie, 1993), and (3) from the theoretical variograms the kriging weights for $n=6$ neighbouring streamgauges are computed

for any prediction location and the streamflow index of interest is predicted at that location. $n$ equal to 6 was set after a preliminary sensitivity analysis which confirmed the results obtained in previous studies (Pugliese et al., 2014, 2016), suggesting to keep the size of nearest neighbours limited when interpolating streamflow indices, and TND values in particular. Moreover, differently to what done for the Danube region in Castellarin et al. (2018), for the application to the entire European continent, 300 km is set as the maximum distance from the prediction location within which gauged basins are included for

the prediction itself.

In this study, TNDTK is used for predicting long-term FDCs at prediction points located at the outlets of 21,664 elementary catchments (elementary catchments minimum drainage area: 0.01 km$^2$; average drainage area: 172.83 km$^2$; maximum drainage area: 1668.38 km$^2$) located within the European continent. In particular, interpolation is performed only within watersheds including at least one measuring point of the DG-JRC dataset (see blue area in Fig. 3), whereas no FDC prediction has been

performed for elementary catchment belonging to watershed without measuring points (see black area in Fig. 3).

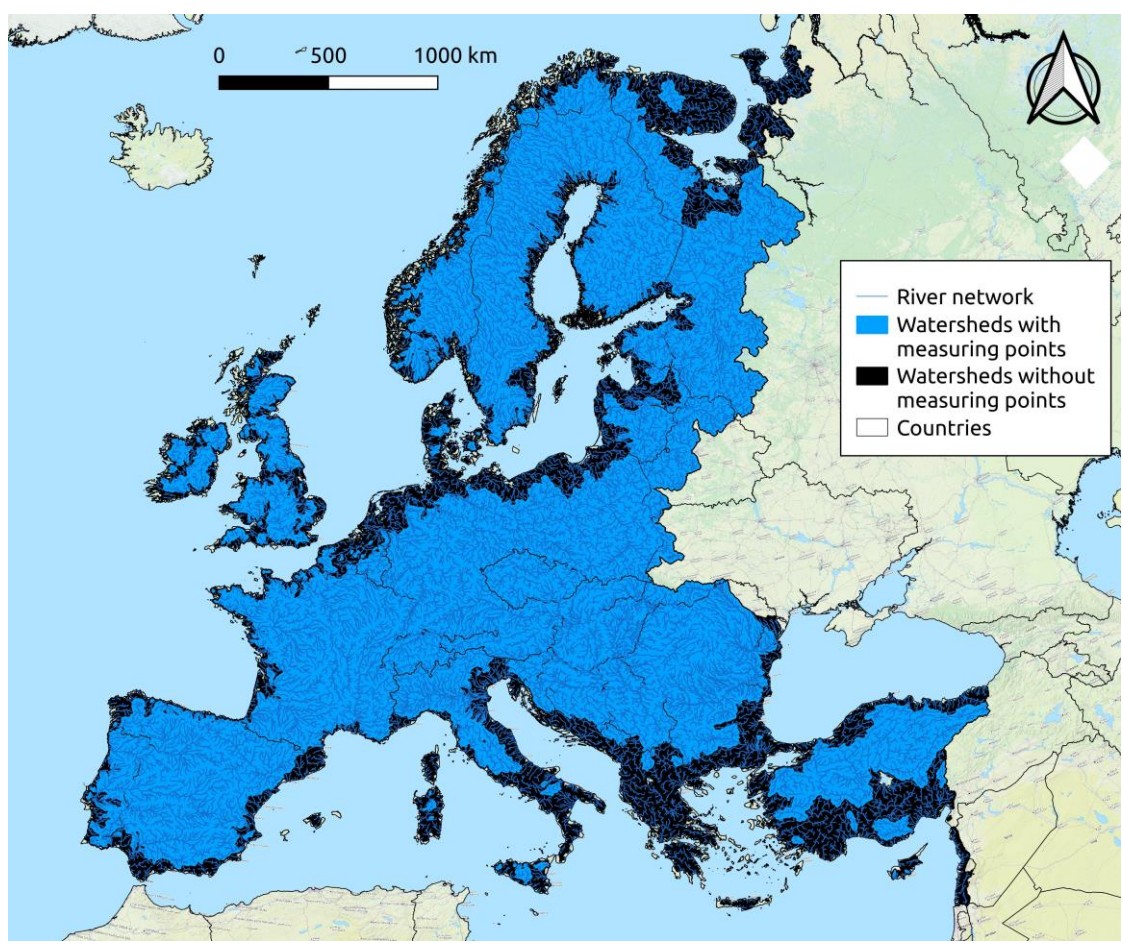

**Figure 3. Merger of the elementary catchments from the DG-JRC database. Predictions have been performed for elementary catchments within the blue area, while no predictions have been provided for the black area.**

## 3.2 Cross-validation

In order to quantitatively test the reliability and robustness of the predicted FDCs, a leave-one-out cross-validation (LOOCV) procedure is used to simulate ungauged conditions at each and every gauged location in the study area. To this aim, the kriging interpolation of MAP and TND values has been performed by adopting a LOOCV strategy (Castellarin et al., 2018; Pugliese et al., 2014). The performance of the proposed model is quantitatively assessed in terms of Nash-Sutcliffe Efficiency between empirical and predicted FDCs by referring to log-transformed streamflows (LNSE) computed either locally (i.e. at each gauge) and globally (i.e. assessing global LNSE values duration-wise). Fig. 4 shows scatter diagrams between empirical and predicted values of MAF, dimensionless and dimensional FDC for the study region obtained by means of the top-kriging interpolation in LOOCV.



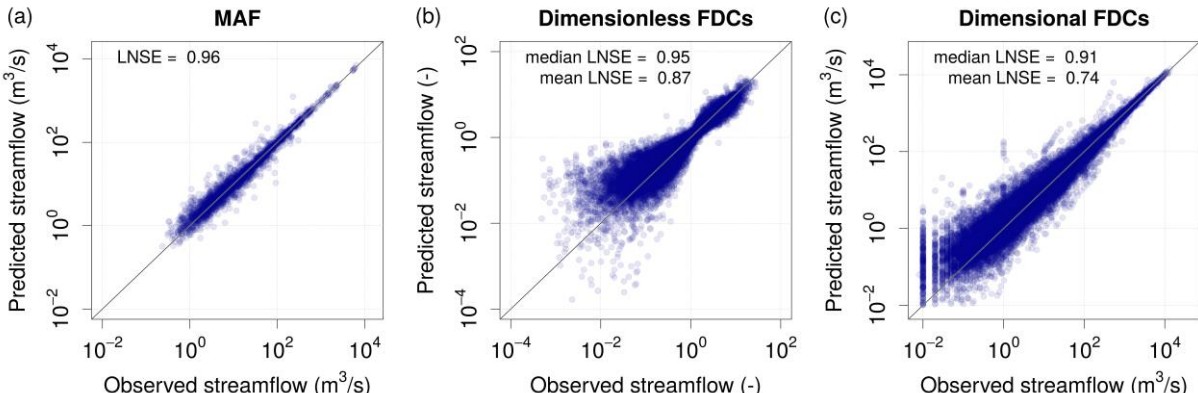

**Figure 4. Results of the top-kriging interpolation in cross-validation (LOOCV). For (a) MAF, (b) dimensionless FCDs, and (c) dimensional FDCs, empirical (x-axes) vs predicted (y-axes) values are reported together with the overall Nash-Sutcliffe efficiency for log-transformed (LNSE) streamflows.**

Fig. 4 highlights the good agreement between observed and predicted values, with LNSE equal to 0.96 for MAF, median and
mean values of at-site LNSEs equal to 0.95 and 0.87 for dimensionless FDCs, and median and mean LNSEs equal to 0.91 and 0.74 for dimensional FDCs. Moreover, the LNSE values computed by comparing duration-wise predicted and empirical streamflow quantiles across all sites for the 15 durations considered in the study (Fig. 5) indicate good performance, with LNSEs well above 0.75 for all durations. A slightly worse performance is observed in the low-flow section of the curves, which is expected (see e.g. Castellarin et al., 2018).

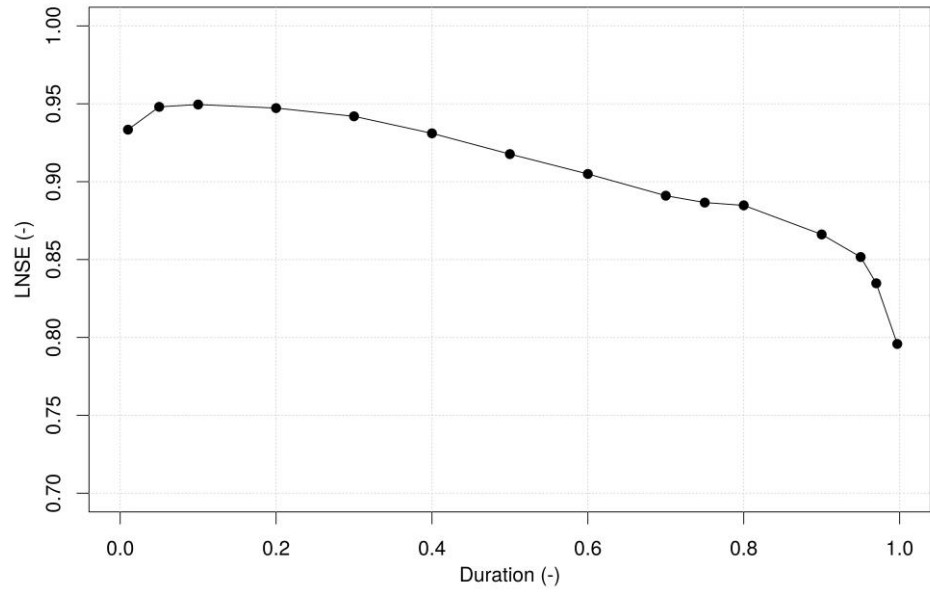


**Figure 5. Cross-validation (LOOCV) of predicted dimensional FDCs. LNSE values are reported as a function of duration.**

### 3.3 Assessment of prediction uncertainty

The above-mentioned outcomes refer to aggregated measures of reliability. Given the wide extension of the study area, it becomes apparent that the TNDTK predictions of streamflow regime are associated with different degrees of reliability, which varies significantly over the study area. Therefore, we develop for the entire study region (i.e. European continent) a measure of uncertainty to be attached to FDC predictions. This measure should guide practitioners and users of interpolated flow-duration curves providing them with an operational tool for judging the suitability of the predicted FDCs for the water-problem at hand. In order to assess prediction uncertainty (i.e. estimate of the interpolation error) one can exploit the predictor variance,

which is an output of top-kriging, as of any kriging-based interpolation procedure (Castellarin et al., 2018). This statistic is a combination of model uncertainty and configuration of observation locations: lower kriging variances are expected for large prediction catchments surrounded by several streamgauges, whereas higher variances are expected for prediction nodes located in data-scarce sub-areas and in upstream catchments. Fig. 6 reports the moving average of standardized prediction variances (i.e. standardization with the maximum value, y axis) as a function of LNSE values (x axes) for a subset of 250 catchments.

Note that for the 2484 high-quality streamgauges, only 2101 LNSE values are available: the presence of measured percentiles of daily flows equal to zero makes the evaluation of LNSE not possible for 382 streamgauges.

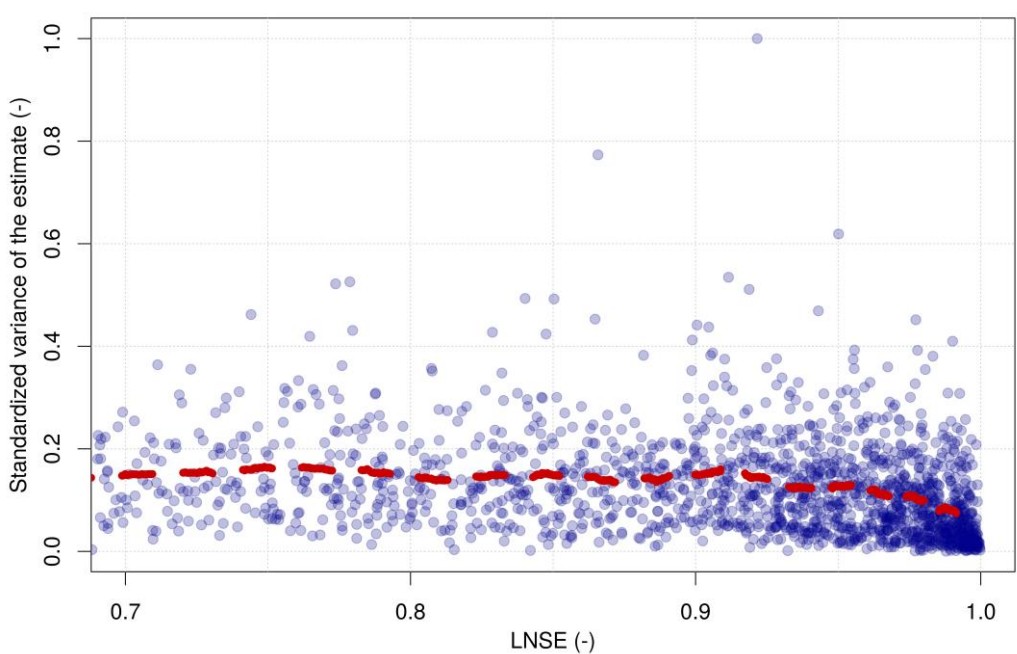

**Figure 6. Standardized prediction variance of TNDTK as a function of LNSE. LNSE values smaller than 0.7 are here omitted. Red dashed line represents the moving average computed with a moving window of 150 catchments. Note that for the 2484 high-quality**
**streamgauges, only 2102 LNSE values are available.**



As observed in Castellarin et al. (2018) for the Danube region, Fig. 6 confirms for the European continent that higher LNSE values are associated with lower prediction variances. This information is useful for the application of TNDTK to ungauged basins since the average density of prediction nodes for the European region (i.e. approximately $5 \cdot 10^{-3}$ prediction points per km$^2$) is significantly higher than the streamgauging network density (i.e. approximately $6 \cdot 10^{-4}$ gauges per km$^2$ for high-quality data).

### 3.4 Qualitative indicator of prediction accuracy

Furthermore, aiming at providing a qualitative assessment of the accuracy of predicted FDCs, we refer to Fig. 6. First of all, we standardize prediction variances with the maximum value (0.28) among gauged and ungauged catchments, and then we compute a threshold value KV* (where KV stands for kriging variance) as the value below which the number of gauged catchments having LNSE<0.8 is equal to 5% of the total amount of gauged stations. Then ungauged catchments having standardized prediction variances lower and higher than KV* are labelled with "yes" and "no", respectively. For gauged catchments LNSE values obtained in LOOCV are attached.

### 4 Data availability

The data produced in this study are freely available at the open access library PANGAEA (Data Publisher for Earth & Environmental Science) at https://doi.pangaea.de/10.1594/PANGAEA.938975 (Persiano et al., 2021). The dataset consists of a GIS vector layer of the contours of 24,148 elementary catchments contained within the white area reported in Fig. 3. The elementary catchments include both the 2484 high-quality gauged catchments and the 21,664 predictions nodes for which FDCs are estimated. Table 1 reports a summary of the information associated with each catchment, including the predicted FDCs' percentiles and the measures of uncertainty and accuracy described above. The file is stored using the ESRI Shapefile format in the ETRS89 (European Terrestrial Reference System 1989) – LAEA (Lambert Azimuthal Equal Area) datum and geographic coordinate system.

**Table 1. Description of fields of the produced GIS layer reporting predicted FDCs for the entire European continent. Note that only the 2484 catchments with high-quality data are considered as gauged and used for predicting FDCs in ungauged locations.**

| Field | Description |
|---|---|
| HydroID | ID code associated with the corresponding elementary catchment |
| NextDownID | ID code associated with the elementary catchment located immediately downstream of the selected one; NextDownID is set to -1 if no elementary catchment is present downstream within the GIS layer |




| Area_EC | Area of the elementary catchment ($km^2$) |
|---|---|
| Area_AC | Area of the whole catchment ($km^2$)<br><br>(for headwater catchments, Area_AC=Area_EC) |
| Gauged | Distinguishes between gauged and ungauged nodes: "yes" for gauged catchments (i.e. 2484 catchments with high-quality data); "no" for ungauged catchments (i.e. 21,664 prediction nodes). |
| Var_est | TNDTK prediction variance for each elementary catchment (-) |
| TND | Empirical value of the total negative deviation, TND (Pugliese et al., 2014) (-) |
| MAF | Mean Annual Flow ($m^3\,s^{-1}$) * |
| D9Q1 | 1st streamflow percentile ($m^3\,s^{-1}$) * |
| D9Q5 | 5th streamflow percentile ($m^3\,s^{-1}$) * |
| D9Q10 | 10th streamflow percentile ($m^3\,s^{-1}$) * |
| D9Q20 | 20th streamflow percentile ($m^3\,s^{-1}$) * |
| D9Q30 | 30th streamflow percentile ($m^3\,s^{-1}$) * |
| D9Q40 | 40th streamflow percentile ($m^3\,s^{-1}$) * |
| D9Q50 | 50th streamflow percentile ($m^3\,s^{-1}$) * |
| D9Q60 | 60th streamflow percentile ($m^3\,s^{-1}$) * |
| D9Q70 | 70th streamflow percentile ($m^3\,s^{-1}$) * |
| D9Q75 | 75th streamflow percentile ($m^3\,s^{-1}$) * |
| D9Q80 | 80th streamflow percentile ($m^3\,s^{-1}$) * |
| D9Q90 | 90th streamflow percentile ($m^3\,s^{-1}$) * |
| D9Q95 | 95th streamflow percentile ($m^3\,s^{-1}$) * |
| D9Q97 | 97th streamflow percentile ($m^3\,s^{-1}$) * |
| D9Q997 | 99.7th streamflow percentile ($m^3\,s^{-1}$) * |
| LNSE | LNSE values obtained in LOOCV for gauged catchments; for ungauged catchments, the distinction is made between catchments having prediction variance lower ("yes") and higher than KV* ("no"). |





| | No values are reported for those gauged catchments where the presence of measured percentiles of daily flows equal to zero makes the evaluation of LNSE not possible. |
|---|---|
| LakeRatio | Ratio between area covered with bodies of water and total area for each elementary catchment (-). |

* If the field "Gauged" is equal to "yes", then the marked values are empirical, while they are predicted otherwise.

## 5 Conclusions

The present study describes an original spatial dataset providing a continuous representation of the streamflow regime for 24,148 elementary river catchments across Europe. For each elementary catchment, the boundaries are provided together with the value of corresponding drainage area (for both elementary and whole catchment), mean annual flow (i.e. long-term average

daily discharge) and a set of 15 streamflow quantiles for duration from 1 to 99.7% (i.e. flow-duration curves, FDCs). The streamflow indices (i.e. mean annual flood and FDCs) have been estimated by means of a geostatistical procedure (i.e. total negative deviation top-kriging; TNDTK) starting from the empirical observations available for about 3,000 discharge measurement stations included in the dataset itself. Consistent with what observed by Castellarin et al. (2018) for the Danube region, the adopted procedure provides an overall good accuracy for the entire study area (i.e. European continent), with a

reliability which can vary significantly in space, depending mainly on streamgauging network density. For this reason, for each catchment we provide indicators of the accuracy and reliability of the performed large-scale geostatistical prediction, i.e. a measure of the prediction performance for gauged elementary catchments and a measure of uncertainty (i.e. estimate of the interpolation error, associated with kriging variance) for the streamflow indices estimated at ungauged catchments. These measures should provide practitioners and users with an operational tool for judging the suitability of the predicted FDCs for

the water-problem at hand. Overall, the dataset made available herein is expected to be useful for the evaluation of water resources availability at ungauged locations, and as a benchmark for the development of hydrological macroscale models.

## Code Availability

All the activities regarding data preparation, analysis and representation of results have been produced by the use of free and open-source software (i.e. Quantum GIS Geographic Information System - Open Source Geospatial Foundation Project,

http://qgis.osgeo.org, and the R Project for Statistical Computing, https://www.R-project.org/). A repository containing an application example for extracting POR-FDCs from daily streamflow series observed at gauged sites and computing FDCs at ungauged target sites by means of total negative deviation top-kriging (TNDTK; Pugliese et al., 2014, 2016) is available at the following link: https://zenodo.org/badge/latestdoi/366670107. The specific R codes used for data preparation and analysis presented in this paper are made available by the authors upon request.





**Acknowledgements**

The source data used for this analysis were compiled by the Joint Research Centre of the European Commission (DG-JRC) from the archives of the International Commission for the Protection of the Danube River (ICPDR) and the Global Runoff Data Centre (GRDC). Both institutions are gratefully acknowledged for providing the data.

**Author contributions**

Simone Persiano wrote the paper and, together with Alessio Pugliese, developed the R scripts. performed the analyses and produced the figures. Alberto Aloe and Jon Olav Skøien provided support regarding the source data used for the analysis and the methodologies. Attilio Castellarin and Alberto Pistocchi contributed to the development of the paper idea and supported the writing process of the manuscript.

**Competing interests**

The authors declare that they have no conflict of interest.

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
