# Peer review of "Streamflow data availability in Europe: a detailed dataset of interpolated flow-duration curves"

_Earth System Science Data, 2021_

## Referee Comment (RC2)

**Review Comments: Streamflow data availability in Europe: a detailed dataset of interpolated flow-duration curves**

Persiano et al have used a statistical procedure named Total Negative Deviation Top-Kriging to estimate flow duration curves for a large sample of river basins in Europe.

I will provide a brief summary of my understanding of the paper. Perhaps the authors would be able to correct me if I am wrong.

2484 stream gauges for which empirical FDCs can be calculated are used as input (training) data, and the target is to produce predicted FDCs for 32,960 nodes across the European river network (ungauged sites). No other information about catchment geology or other catchment characteristics is used as a covariate/independent variable, just the FDCs for the 2484 training gauges.

The top-kriging approach is applied and results are presented for all 32,960 prediction nodes (i.e. in Fig 4 the blue points are individual prediction nodes). Performance scores are estimated using a leave-one-out cross validation procedure to produce robust estimates of performance of the method.

Uncertainty is estimated using the predictor variance of the top-kriging estimator. This shows that as the LNSE of the training (gauged) sites increases, the predictor variance declines. This therefore shows that more accurate estimates are less uncertain (smaller variance).

The paper is well written and concise, and I commend the authors for the length of their manuscript. It was well received.

I believe that the manuscript can be recommended for publication subject to minor corrections.

**Specific Comments:**

**L106--110**: This may be naive, but based on Figure 1 in Pugliese et al 2014 (screenshotted below), it seems to me as if TND does not describe the whole curve, it captures the majority but fails to capture the low duration flood events (high flows). Is that an important aspect of the method? Does that have any impact on the estimation of FDCs?

[Figure]

**Figure 1.** Total negative deviation (TND, filled area) for three catchments with different hydrological behaviours (see Sect. 4). Top panels: $TND_1$ (red area) for an empirical FDC (black thick line) standardised by mean annual flow (MAF); bottom panels: $TND_2$ (blue area) for an empirical FDC (black tick line) standardised by $MAP^* = MAP \cdot A \cdot CF$, where MAP is the mean annual precipitation, $A$ is the drainage area, and CF is a unit-conversion factor.

**L143–144 & Fig. 3**:

**L16 & L205**: It is claimed that the data layer is available, however I had difficulty accessing the data on Pangea: "*Download Data (login required; moratorium until 2023-12-03)*". Does this mean that we are unable to access the data until the end of 2023?

**Download Data (login required; moratorium until 2023-12-03)**

Download dataset

**L144-145**: "*In particular, interpolation is performed only within watersheds including at least one measuring point of the DG-JRC dataset (see blue area in Fig. 3), whereas no FDC prediction has been performed for elementary catchment belonging to watershed without measuring points (see black area in Fig. 3).*"

The final portion of the sentence is a little unclear. I would suggest rewriting:
"*In particular, interpolation is performed only within watersheds including at least one measuring point of the DG-JRC dataset (see blue area in Fig. 3). Watersheds where there are no measuring gauges are excluding from our analysis (see black area in Fig. 3).*"

**Fig 4 & L164–169**: The results do look good! One thing that I would appreciate some insight into is the large difference between the Median and Mean NSE, this suggests highly skewed performances with long lower-LNSE tails.

Would it be possible to see a histogram of these performances? Or at least a comment on the characteristics of the catchments with worse performing predictions? You have mentioned that the performance is worse for lower-flow sections, so does that mean that for dryer (more arid) catchments the performance is worse?

**L233-240:** If possible, I believe that it would also be great for the community to be able to test the procedure, perhaps in a Rmarkdown or Jupyter notebook form. I believe that this would add extra weight to the paper, allowing for full reproducibility and transparency of the method. Does the code found here: https://github.com/SimonePersiano/TNDTK/tree/v1.0.0 provide a sufficient overview to reproduce the TNDK method for ourselves?

**L230-231**: Would it be possible to provide some further context of how this dataset might be used as a "benchmark for the development of hydrological macroscale models"?

---

## Author Response (AR1)

We would like to thank Anonymous Referee #1 for his/her comments and suggestions: we believe they helped us in improving the presentation of our study. For easing the reading of our rebuttal, original Reviewers Assessment is reported in italics (tag "Reviewer"), while our replies are flagged with the tag "Authors".

1. **Reviewer**: *This is a brief and well-written manuscript describing a data set of statistically estimated flow duration curves (at a density of 15 quantiles) for basins across the European continent. The manuscript uses existing methods and cites them appropriately, leading to a thoroughly described methodology. I have no concerns with eventual publication, and I congratulate the authors on a succinct and thorough work.*

   **Authors**: We thank Anonymous Referee #1 for the positive feedback.

2. **Reviewer**: *If space allows, I would like the authors to opine a bit more on this statement: "An empirical characterization of the natural streamflow regime over large areas would be a fundamental piece of information for benchmarking the performance of macro-scale models and for assessing their potential locally" (lines 33-34). I certainly agree, and I find it to be a motivating factor for this work. I was disappointed not to see it strongly revisited in the discussion or conclusions. Perhaps it is beyond the scope of a data manuscript, but I am curious about how the authors see this work advancing that goal. How would an estimated FDC, with inherent uncertainty, be used to train a macro model? How would you account for the uncertainty? What would be the implications of this uncertainty? There are, of course, no definitive answers here, but a brief discussion may deepen the impact of this work.*

   **Authors**: We agree with Anonymous Referee #1 (and Anonymous Referee #2, who raised the same issue) on the need to give more room to the mentioned points in the manuscript. The local performances of macro-scale models are highly variable, depending on the quality of macro-scale input data and the adequacy of the conceptual scheme to represent the hydrological processes that locally drive the rainfall-runoff transformation. On the other hand, TNDTK relies on a small amount of input data (i.e., streamflow series and catchments' size and mutual position; see e.g. Pugliese et al., 2016) for predicting the natural streamflow regime (i.e., FDCs) over large areas. Based on this, Pugliese et al. (2018) showed that TNDTK can be a useful tool for enhancing results from macro-scale models along the stream network of a given region, with significant advantages even for very low stream-gauging network densities. Indeed, being a geostatistical model, TNDTK is theoretically unbiased (i.e., BLUE procedure) and its FDC predictions can be effectively employed to correct systematic bias associated with the outcomes of macro-scale rainfall-runoff models. The enhancement procedure, which is thoroughly described in Pugliese et al. (2018) (please see the reference already included in the original version of the manuscript), is based on the comparison of the TNDTK-estimated FDCs with the rainfall-runoff simulations (i.e., macro-scale model) and the computation of a so-called residual-duration curve to enhance macro-scale simulated streamflows. Such a procedure does not explicitly account for the uncertainty associated with the predicted FDC, yet TNDTK returns a prediction variance, that is an estimate of the model uncertainty (i.e., combination of model uncertainty and configuration of observation locations) and therefore can be used as guidance for benchmarking and enhancing macro-scale models. The field "Var_est" in our published dataset represents TNDTK prediction variance for each elementary catchment and can be used to this aim. Of course, we agree with Anonymous Referee #1 that future studies should address

the important issue of including prediction variance for benchmarking and enhancing purposes. In the revised manuscript, we expanded the Conclusions section to include the above-mentioned considerations on the ready usage of the dataset for hydrological applications.

**Anonymous Referee #2**

Many thanks to Anonymous Referee #2 for his/her comments and suggestions, that are strongly appreciated. We believe that they significantly contributed to the improvement of our manuscript. We propose to address the comments as we illustrate below. For easing the reading of our rebuttal, original Reviewers Assessment is reported in italics (tag "Reviewer"), while our replies are flagged with the tag "Authors".

**General Comments:**

1. **Reviewer**: *Persiano et al have used a statistical procedure named Total Negative Deviation Top-Kriging to estimate flow duration curves for a large sample of river basins in Europe.*
   *I will provide a brief summary of my understanding of the paper. Perhaps the authors would be able to correct me if I am wrong. 2484 stream gauges for which empirical FDCs can be calculated are used as input (training) data, and the target is to produce predicted FDCs for 32,960 nodes across the European river network (ungauged sites). No other information about catchment geology or other catchment characteristics is used as a covariate/independent variable, just the FDCs for the 2484 training gauges. The top-kriging approach is applied and results are presented for all 32,960 prediction nodes (i.e. in Fig 4 the blue points are individual prediction nodes).*

   **Authors**: The brief summary provided by Anonymous Referee #2 is correct. We would just point out that the blue points in Fig. 4 represent the 15 FDCs' quantiles predicted in cross-validation for each of the 2484 streamgauges (where empirical FDCs were available).

2. **Reviewer**: *Performance scores are estimated using a leave-one-out cross validation procedure to produce robust estimates of performance of the method. Uncertainty is estimated using the predictor variance of the top-kriging estimator. This shows that as the LNSE of the training (gauged) sites increases, the predictor variance declines. This therefore shows that more accurate estimates are less uncertain (smaller variance).*
   *The paper is well written and concise, and I commend the authors for the length of their manuscript. It was well received. I believe that the manuscript can be recommended for publication subject to minor corrections.*

   **Authors**: We thank Anonymous Referee #2 for the positive comments.

**Specific Comments**:

3. **Reviewer**: *L106-110: This may be naive, but based on Figure 1 in Pugliese et al 2014 (screenshotted below), it seems to me as if TND does not describe the whole curve, it captures the majority but fails to capture the low duration flood events (high flows). Is that an important aspect of the method? Does that have any impact on the estimation of FDCs?*

**Authors**: We thank Anonymous Referee #2 for this comment, which allows us to better clarify some important aspects of the procedure we adopted to produce the dataset. Total negative deviation (TND) was first introduced by Pugliese et al. (2014) as a point-index useful for representing the shape of the FDC and, in this way, capturing the main characteristics of the catchment. Indeed, even though TND does not describe the portion of the FDC associated with low durations (high flows), it is very informative on the shape of the FDC, that is controlled by climatic, physiographic, and geo-pedological characteristics of the catchment (see Pugliese et al., 2014). Larger TNDs are associated with steeper FDCs (i.e., catchments with rapidly responding runoff processes), while smaller TNDs are associated with less steep FDCs. Therefore, thanks to its capability to express the hydrological similarity between catchments, TND can be effectively used to derive the geostatistical weights within the TDNTK procedure. On the other hand, FDC predictions are performed by using empirical standardized FDCs as a whole, so that the entire duration interval is considered, low durations included. As a result, though worse prediction performance can be observed in some cases for low and very-high flows, their overall accuracy still remains good (see Figure 5 of the original version of our manuscript, as well as previous studies, e.g., Castellarin et al., 2018), unless prediction is performed in areas with low gauging density. We added these considerations in the revised version of the manuscript.

4. **Reviewer**: *L16 & L205: It is claimed that the data layer is available, however I had difficulty accessing the data on Pangea: "Download Data (login required; moratorium until 2023-12-03)". Does this mean that we are unable to access the data until the end of 2023?*

   **Authors**: We thank the Reviewer for highlighting such an important issue. At the time of submission, metadata were publicly accessible, yet we decided to protect the access to data until this publication was officially out. ESSD Reviewers could have been able to access it through a temporary key, which we communicated to the Editors but which we erroneously did not explicitly declare in the manuscript. Since the temporary key has expired now, we have decided to remove the moratorium and make data publicly accessible. The dataset is available at the "Download Data" section at the same link already included in the original version of the manuscript: https://doi.pangaea.de/10.1594/PANGAEA.938975.

5. **Reviewer**: *L144-145: "In particular, interpolation is performed only within watersheds including at least one measuring point of the DG-JRC dataset (see blue area in Fig. 3), whereas no FDC prediction has been performed for elementary catchment belonging to watershed without measuring points (see black area in Fig. 3)."*
   *The final portion of the sentence is a little unclear. I would suggest rewriting: "In particular, interpolation is performed only within watersheds including at least one measuring point of the DG-JRC dataset (see blue area in Fig. 3). Watersheds where there are no measuring gauges are excluding from our analysis (see black area in Fig. 3)."*

   **Authors**: We thank Anonymous Referee #2 for pointing this out. We rephrased as follows: "In particular, interpolation is performed only within watersheds including at least one measuring point of the DG-JRC dataset (see blue area in Fig. 3). Elementary catchments within watersheds where no measuring gauges are present have been excluded from the analysis (see black area in Fig. 3)."

6.  **Reviewer**: *Fig 4 & L164–169: The results do look good! One thing that I would appreciate some insight into is the large difference between the Median and Mean NSE, this suggests highly skewed performances with long lower-LNSE tails.*
    *Would it be possible to see a histogram of these performances? Or at least a comment on the characteristics of the catchments with worse performing predictions? You have mentioned that the performance is worse for lower-flow sections, so does that mean that for dryer (more arid) catchments the performance is worse?*

    **Authors**: Many thanks for the comment, that allows us to better clarify the different expected performance of the model depending on catchment characteristics (i.e., drainage area, mutual position of catchments, nested structure, climate; see also Pugliese et al., 2016). As correctly pointed out by Anonymous Referee #2, the prediction of both dimensionless and dimensional FDCs (see panels (b) and (c) in Figure 4, respectively) is associated with a mean value of LNSE which is lower than its median. This behaviour is expected and can be explained by the presence of some sites characterized by extremely low prediction performance. As shown in the application of TNDTK within the Danube river basin published in Castellarin et al. (2018), TNDTK tends to perform better for larger catchments, whereas lower performance is expected for smaller catchments, especially for headwater ones. In general, the combination of small catchment areas and wide climatic variability may affect TNDTK ability to predict variation in streamflow regimes, especially regarding very-high durations (i.e. severe droughts). As a result, TNDTK tends to overestimate low flows (see Pugliese et al. 2016). Also, rather poor predictions can be expected for catchments characterized by a severe anthropic alteration of streamflow regime (e.g., river sections downstream lakes and reservoirs). We added the above-mentioned considerations in the revised version of the manuscript. For a thorough description of the main causes of worst-case performances, the reader has been referred to the above-mentioned studies (e.g., Pugliese et al, 2016; Castellarin et al., 2018).

7.  **Reviewer**: *L233-240: If possible, I believe that it would also be great for the community to be able to test the procedure, perhaps in a Rmarkdown or Jupyter notebook form. I believe that this would add extra weight to the paper, allowing for full reproducibility and transparency of the method. Does the code found here: https://github.com/SimonePersiano/TNDTK/tree/v1.0.0 provide a sufficient overview to reproduce the TNDK method for ourselves?*

    **Authors**: We are grateful to Anonymous Referee #2 for the valuable suggestion. We believe that the codes published in the Github provide sufficient information for reproducing the TNDTK method, yet we agree with the Reviewer that allowing one to test the procedure would be useful for the community. To this aim, we have prepared an R-Markdown notebook, where the codes included in the Github are commented step-by-step. Since the application of the adopted procedure to the entire European study area requires a non-negligible amount of RAM and computational times, both the Github (Zenodo) and the R-Markdown refer to an example application including 30 catchments located in Tyrol (Austria) and South Tyrol (Italy) (see also the "Code Availability" section of the original manuscript). The R-Markdown notebook has been published on RPubs (see https://rpubs.com/) and the link has been included in the Code Availability section of the revised version of the manuscript.

8.  **Reviewer**: *L230-231: Would it be possible to provide some further context of how this dataset might be used as a "benchmark for the development of hydrological macroscale models"?*

**Authors**: We acknowledge Anonymous Referee #2 for this comment, which is in line with an analogous issue raised by Anonymous Referee #1. As we wrote in our reply to Anonymous Referee #1, we provided more room to the usefulness of the dataset as a benchmark for the development of hydrological macroscale models (for more details, please see point 2. in our reply to Anonymous Referee #1).